# Healthy Eating and Mortality among Breast Cancer Survivors: A Systematic Review and Meta-Analysis of Cohort Studies

**DOI:** 10.3390/ijerph19137579

**Published:** 2022-06-21

**Authors:** Eunkyung Lee, Vanessa Kady, Eric Han, Kayla Montan, Marjona Normuminova, Michael J. Rovito

**Affiliations:** Department of Health Sciences, College of Health Professions and Sciences, University of Central Florida, 4364 Scorpius Street, Orlando, FL 32816, USA; vanessa_kady@knights.ucf.edu (V.K.); ericmhan@knights.ucf.edu (E.H.); kmontan@knights.ucf.edu (K.M.); marjonanormuminova@knights.ucf.edu (M.N.); michael.rovito@ucf.edu (M.J.R.)

**Keywords:** dietary guidelines, diet quality, breast cancer mortality, Diet Approaches to Stop Hypertension, systematic review, meta-analysis

## Abstract

This systematic review examined the effect of diet quality, defined as adherence to healthy dietary recommendations, on all-cause and breast cancer-specific mortality. Web of Science, Medline, CINAHL, and PsycINFO databases were searched to identify eligible studies published by May 2021. We used a random-effects model meta-analysis in two different approaches to estimate pooled hazard ratio (HR) and 95% confidence interval (CI) for highest and lowest categories of diet quality: (1) each dietary quality index as the unit of analysis and (2) cohort as the unit of analysis. Heterogeneity was examined using Cochran’s Q test and inconsistency I^2^ statistics. The risk of bias was assessed by the Newcastle–Ottawa Scale for cohort studies, and the quality of evidence was investigated by the GRADE tool. The analysis included 11 publications from eight cohorts, including data from 27,346 survivors and seven dietary indices. Both approaches yielded a similar effect size, but cohort-based analysis had a wider CI. Pre-diagnosis diet quality was not associated with both outcomes. However, better post-diagnosis diet quality significantly reduced all-cause mortality by 21% (HR = 0.79, 95% CI = 0.70, 0.89, I^2^ = 16.83%, *n* = 7) and marginally reduced breast cancer-specific mortality by 15% (HR = 0.85, 95% CI = 0.62, 1.18, I^2^ = 57.4%, *n* = 7). Subgroup analysis showed that adhering to the Diet Approaches to Stop Hypertension and Chinese Food Pagoda guidelines could reduce breast cancer-specific mortality. Such reduction could be larger for older people, physically fit individuals, and women with estrogen receptor-positive, progesterone receptor-negative, or human epidermal growth factor receptor 2-positive tumors. The risk of bias in the selected studies was low, and the quality of evidence for the identified associations was low or very low due to imprecision of effect estimation, inconsistent results, and publication bias. More research is needed to precisely estimate the effect of diet quality on mortality. Healthcare providers can encourage breast cancer survivors to comply with healthy dietary recommendations to improve overall health. (Funding: University of Central Florida Office of Undergraduate Research, Registration: PROSPERO-CRD42021260135).

## 1. Introduction

In 2020, approximately 2.3 million women were diagnosed with breast cancer, accounting for 11.7% of all new cancer cases worldwide, and 685,000 women died from breast cancer, accounting for 6.9% of all cancer deaths [1]. The disease is transformative for the patient, not just physiologically but also socio-behaviorally, with a particular focus on diet. Cancer survivors tend to change their food choices following a cancer diagnosis, hoping to positively influence their prognosis [2]. Many epidemiologic studies have shown that poor diet quality, such as the Healthy Eating Index (HEI), is strongly and positively associated with obesity and a higher body mass index (BMI) [3,4], a major risk factor for several non-communicable diseases, including cardiovascular diseases and many cancers [5,6]. Poor diet quality also has been associated with higher serum proinflammatory cytokines [7,8] and breast density [9,10,11], which are associated with cancer risk [12]. Another mechanism for a healthy diet that can lead to better cancer survival could include controlling tumor promotors through improved weight loss and insulin sensitivity with a better diet [13,14].

Sun et al. [15] showed that 28% of breast cancer survivors from the participants in the Women’s Health Initiative (WHI) study in the United States changed their diet quality after a cancer diagnosis, and Thompson et al. [16] reported increased vegetable and fruit consumption and decreased dietary fat consumption after a cancer diagnosis among women treated for invasive breast cancer in the United States. However, not all of these dietary changes are necessarily healthier in the long run. Sun et al. [15] explained that 9% of their sample had a decrease in diet quality. Furthermore, Lee et al. [17] reported an average HEI-2015 score of 55.6 (out of 100) among American adult cancer survivors indicating that there could still be room for improvement. Another study reported that nearly half of those with cancer had tried special diets [18], which may have adverse effects on their health if followed for long periods of time. One reason behind this heterogeneity effect may be that there are no dietary guidelines specifically for cancer survivors [19]. Although the American Cancer Society (ACS) [20] and the World Cancer Research Fund/American Institute of Cancer Research (WCRF/AICR) [21,22,23] published lifestyle recommendations for cancer prevention, there is still a complete need for dietary recommendations specific to cancer survivors. Considering the above evidence that patient-driven dietary changes may not always lead to a better diet, there is a pressing need for informed guidance for them.

To these suggestions, researchers examined the effect of adhering to the dietary recommendations for overall health on cancer outcomes. Multiple meta-analysis studies showed that following a healthy diet improved overall survival among cancer survivors, including breast cancer. In 2020, Morze et al. [24] conducted the second update on their systematic review and meta-analysis and concluded that better diet quality could reduce the risk for all-cause death by 17% and cancer death by 18%. However, they did not report results separately for breast cancer survivors. In 2016, Schwedhelm et al. [25] examined the effect of diet quality, measured by adherence scores to Mediterranean diet (MED), Dietary Approach to Stop Hypertension (DASH), HEI/altered version of HEI (AHEI), and WCRF/AICR guidelines, upon on all-cause mortality from 117 studies. Out of these 117 studies included in their review, only three studies [26,27,28] pertained to breast cancer survivors, and a meta-analysis of these three studies showed a 26% reduced risk for all-cause death (RR = 0.74, 95% CI = 0.60, 0.90) [25]. However, there was insufficient evidence regarding breast cancer-specific death and recurrence due to limited numbers of studies for those outcomes.

In 2017, Schwingshackl et al. [29] updated their meta-analysis, focusing on only HEI/AHEI/DASH scores. Their review included 68 studies with only 2 studies [26,28] reporting results for breast cancer survivors. Although the findings were not significant, they indicated that following healthy dietary recommendations could reduce the risk for breast cancer-specific death by 6%. However, they counted one study multiple times when it included multiple dietary indices, which might have biased the estimation. Pourmasoumi and colleagues [30] reported in 2016 that there was no significant association between HEI/AHEI scores and breast cancer-specific survival from four studies [26,27,28,31] (RR = 1.04, 95%CI = 0.69, 1.56) included in their meta-analysis. Similar to the review mentioned above, Pourmasoumi et al. included two studies [27,31] originating from the same cohort, the Nurses’ Health Study, which could lead to a biased summary measure. Special attention is needed for meta-analysis when multiple dietary indices are evaluated in one study or when multiple publications are produced from the same cohort.

Recently, seven more studies [15,32,33,34,35,36,37] examined the associations between diet quality and breast cancer-specific outcomes, allowing re-evaluation of such associations with increased statistical power. Therefore, this current meta-analysis aims (1) to examine the associations between diet quality indices/scores and cancer outcomes (i.e., recurrence and mortality) using two approaches: (i) a dietary index as the unit of analysis (index-based analysis) and (ii) a cohort as the unit of analysis (cohort-based analysis) and (2) to examine whether such associations vary according to cancer subtype or participant characteristics. Employing a cohort-based meta-analysis rather than an index-based analysis will produce a more valid estimate of these measures. In addition, including all available studies will provide sufficient statistical power to identify a diet quality index that has the most favorable impact on breast cancer outcomes, and subgroup analysis will identify target populations for interventions.

## 2. Materials and Methods

The Preferred Reporting Items for Systematic Reviews and Meta-Analyses (PRISMA) statement was used to structure the present study. The review protocol was registered in the International Prospective Register of Systematic Reviews (PROSPERO; University of York; York, United Kingdom), an open-access online database of prospectively registered systematic reviews in health and social care (CRD42021260135).

Two independent reviewers conducted a comprehensive search using the key search terms to maximize the identification of cohort studies that examined the associations between diet quality and breast cancer recurrence and mortality. Web of Science and EBSCOhost (Medline, CINAHL, and PsycINFO) online search platforms were searched on 2 June 2021, using the following search parameters: breast cancer, dietary quality, recurrence, mortality, and prognosis. The full search strategies are shown in Appendix A. Filters included human studies, language (English, Korean, and Spanish), and articles that had been published in a peer-reviewed journal up to 31 May 2021. Abstracts were available in English although the full texts were in Korean or Spanish. Translation of Korean or Spanish to English was planned when the publication was to pass the initial title/abstract screening phase, but none passed the screening. We also reviewed the reference lists of the included studies to identify additional reports that could potentially be eligible.

The search results from all databases were first uploaded to Covidence^®^ software (Veritas Health Innovation, Melbourne, Australia) [38], and inclusion and exclusion criteria were inputted onto the system, which the reviewers could view while screening the studies. Covidence^®^ recognizes the duplicated records from search results and removes them automatically. It also streamlines the screening and data extraction processes by having all search results and selection criteria in one place. By recording responses from two reviewers independently and simultaneously, it increases the study selection’s efficiency and validity. All reviewers were trained to use the program before starting the screening. After each study’s title and abstract were screened, the full text was screened for inclusion. Each study was reviewed by two reviewers and consensus was research. Studies were included in the review if they met the criteria stated in Table 1 and presented the effect estimates (such as hazard ratio [HR] or risk ratio [RR]) with 95% confidence intervals (CIs) for the association between the dietary quality index/score and cancer outcomes.

The quality of the studies was assessed using the modified Newcastle–Ottawa Scale for cohort studies [39]. This tool evaluates the quality of cohort studies on the following three domains: the adequacy of the recruitment and selection of study participants, comparability of comparison groups, and ascertainment of exposure and outcomes. Studies that received a score of 6 or above were considered as high quality. The result of the quality assessment is included in Table 2 showing most of them having a score of 6 or above, and the detailed item scores are presented in Appendix A. The GRADE (Grading of Recommendations, Assessment, Development and Evaluations) tool [40] was used to grade the quality of evidence for association between diet quality and the predefined outcomes. In brief, the quality of evidence from cohort studies starts at ‘low’ quality, and the quality of the evidence is increased or decreased for the following reasons: risk of bias, inconsistency, indirectness, imprecision, publication bias, effect size, plausible confounding, and dose response [41]. Separate judgements on quality of evidence were made for overall pre-diagnosis and post-diagnosis diet quality.

Two reviewers extracted the data using a predesigned data extraction form created by authors on Covidence^®^. Disagreements were infrequent and were resolved through discussion by consensus with all authors. The following data were extracted from each study:Study characteristics: title, first author, year of publication, country of study, cohort name, study design, sample size, study aim, and follow-up periods.Population characteristics: age, race/ethnicity, smoking, body mass index, menopausal status, and tumor characteristics (stage and estrogen receptor (ER) status).Exposure: a diet quality index used, the dietary assessment tools (food frequency questionnaire, dietary record, or dietary recall) and the dietary assessment timing regarding breast cancer diagnosis (before/at or after diagnosis) and target of diet assessed (pre-diagnostic diet or post-diagnostic diet).Comparison: high versus low dietary quality score.Outcome: recurrence and mortality (all-cause, cancer, and noncancer-specific), ascertainment methods (self-report, medical records, vitality records, National Death Index, or death certificate), RR/HR and 95% CI comparing high vs. low index score, covariates included in the multivariable model, and overall findings from the study.

After extraction, the principal study author was contacted if the original publication had missing information essential to the current study.

The meta-analysis was performed by combining the multivariable-adjusted HR/RR of the association between the diet quality score and cancer outcomes from each study using the DerSimonian–Laird random-effects pooling model. We employed two approaches in a meta-analysis. (1) Index-based analysis: the overall effect of a healthy diet on cancer outcomes was summarized by pooling the results from each diet quality index/score using an index as the unit of analysis. (2) Cohort-based analysis: the overall effect was summarized with a cohort being the unit of analysis. If one study evaluated multiple diet quality indices, an average effect from all indices was first estimated using a random-effects model. If numerous publications came from the same cohort using a different diet quality index, the summary effect for the cohort combining all dietary indices across multiple publications was obtained before the meta-analysis.

Subgroup analysis was conducted to investigate potential sources of heterogeneity. First, the effect of the timing of the dietary assessment was assessed to determine whether pre-and post-diagnosis diet quality had a differential impact on cancer outcomes. Second, the effect of each diet quality index was evaluated to identify which dietary quality index had the most favorable impact on cancer outcomes. Third, patient characteristics and tumor characteristics were evaluated to identify the subgroup of women who could benefit more than others. Heterogeneity between studies was quantitatively assessed by the Cochran *Q* test and inconsistency I^2^ test. Sensitivity analyses were conducted, excluding one study at a time to clarify whether the results were robust or sensitive to the influence of a single study. Statistical analyses were performed using the Comprehensive Meta-Analysis software (version 3, Biostat Inc., Englewood, NJ, USA) and R (version 4.1.2) with the metafor package, and a two-tailed *p* < 0.05 was considered statistically significant.

## 3. Results

### 3.1. Results of the Search

Overall, we screened a total of 2471 abstracts, 33 of which were reviewed in full for eligibility. We excluded 22 full-text publications, leaving 11 for data extraction and quality assessment. The detailed steps of the systematic search and selection process are shown in Figure 1 of a PRISMA flow chart. Appendix A provides a list of full-text publications that were excluded, with reasons.

The characteristics of included studies are summarized in Table 2. Most of the studies were conducted using prospective (*n* = 8) [15,26,27,28,31,35,36,37] or retrospective (*n* = 3) [32,33,34] cohort study designs with a mean/median follow-up period of 6–17.2 years, allowing a sufficient time for outcomes to occur, and most studies had excellent follow-up rates upwards of 95%. Nine studies were conducted in the United States, one [34] from Italy, and one [35] from China. The sample size varied between 110 breast cancer survivors in the National Health and Nutrition Examination Survey (NHANES) III study [33] to 4452 in the Cancer Prevention Study (CPS)-II Nutrition Cohort study. [37] Two studies from NHANES III [32,33] and Wang’s Chinese study [35] included women who survived at least five years after diagnosis and counted events only occurring five years post-diagnosis.

#### 3.1.1. Diet Quality Index

Dietary intake was assessed before/at (*n* = 3) [15,34,37] or after (*n* = 10) [15,26,27,28,31,32,33,35,36,37] breast cancer diagnosis using the food frequency questionnaire (*n* = 9) [15,26,27,28,31,34,35,36,37] or 24-h recall (*n* = 2) [32,33] methods. Diet quality pre-diagnosis (*n* = 3) [15,34,37], post-diagnosis (*n* = 10) [15,26,27,28,31,32,33,35,36,37], and change (*n* = 1) [15] was assessed using seven dietary indices/scores measuring the compliance to a priori defined healthy dietary recommendations. Nine studies used the HEI/AHEI, [15,26,27,28,31,32,33,35,36], which measures the compliance to the American Dietary Guidelines. The adherence score to the DASH guideline was evaluated in three studies [27,35,36] and the MED score (MDS) in three studies [33,34,36]. Two studies evaluated the adherence score to ACS dietary guidelines [36,37] and one study used the compliance score to Chinese Food Pagoda guidelines (CHFP, version 2007 or 2016) [35]. In addition, the Diet Quality Index Revised (DQIR) and the Recommended Food Score (RFS) were assessed in one study [31].

Food components included in each diet quality index are summarized in Table 3. These diet quality indices share several core similarities regarding food groups or dietary components. All indices emphasize high intakes of fruit and vegetables and whole grains, and limited intakes of red and processed meats and saturated fats.

#### 3.1.2. Main Outcomes

The disease recurrence was ascertained from the self-reported questionnaire or medical records, and deaths were confirmed from family members, postal service, or linking to the National Death Index or state/national vital status data. Causes of death were examined from the death certificate. Overall, there were 727 breast cancer recurrences and 5001 deaths from all causes, including 2472 breast cancer deaths among 27,346 breast cancer survivors. A total of three studies [27,35,36] examined breast cancer recurrences, while all 11 studies investigated all-cause mortality, and 10 studies [15,26,27,28,31,32,34,35,36,37] assessed breast cancer-specific mortality.

### 3.2. Effects on Breast Cancer Recurrence

Three studies [27,35,36] reported breast cancer recurrence as the primary outcome of their research. Ergas’ study [36] reported no significant association between post-diagnosis diet quality scores and the risk of breast cancer recurrence regardless of the diet quality index used (combined effect of four indices: HR = 1.13, 95% CI = 0.82, 1.55). Izano et al. [27] mentioned no association between the two factors without reporting specific numeric results. Wang and colleagues [35] reported the significant effect of high diet quality scores on a lower risk of breast cancer events, including recurrences, metastases, and deaths (combined effect of four indices: HR = 0.88, 95% CI = 0.78, 0.99). However, they did not report the effect size for each outcome separately due to the low incidence rates of recurrence and metastasis, which hindered further meta-analysis for breast cancer recurrence in our study.

### 3.3. Effects on All-Cause Mortality

Three studies [15,34,37] reported the effect of pre-diagnosis diet quality on all-cause mortality, while ten studies [15,26,27,28,31,32,33,35,36,37] examined the effect of post-diagnosis diet quality. When both pre- and post-diagnosis diet quality scores were included in the analysis with each index as the unit of analysis, adherence to a high-quality diet was significantly and inversely associated with all-cause mortality (HR = 0.80, 95% CI = 0.76, 0.85, I^2^ = 4.5%, *n* = 24). This effect remained consistent for post-diagnosis diet quality scores (HR = 0.78, 95% CI = 0.73, 0.83, I^2^ = 0.0%, *n* = 21), but not for pre-diagnosis diet quality scores (HR = 0.88, 95% CI = 0.73, 1.06, I^2^ = 58.4%, *n* = 3) as shown in Figure 2.

As the results from Kim et al. [31] and Izano et al. [27] studies came from the NHS, George’s [28] and Sun’s [15] results from the WHI, and Deshmukh’s [32] and Karavasiloglou’s [33] results from the NHANES III, while Ergas et al. evaluated four dietary indices from the Pathways Study, and Wang’s study assessed four indices from the Chinese Shanghai Breast Cancer Survival Study (SBCSS), we conducted a meta-analysis using each cohort as the unit of analysis. The cohort-based meta-analysis showed an effect similar to the index-based analysis but with a wider CI for post-diagnosis diet quality, as shown in Figure 3a (HR = 0.78, 95% CI = 0.69, 0.89, I^2^ = 16.83%, *n* = 7). The effect of pre-diagnosis was not significantly associated with all-cause mortality (HR = 0.88, 95% CI = 0.73, 1.06, I^2^ = 59.0%, *n* = 3).

### 3.4. Effects on Breast Cancer-Specific Mortality

Three studies [15,34,37] reported the effect of pre-diagnosis diet quality, and nine studies [15,26,27,28,31,32,35,36,37] examined the effect of post-diagnosis diet quality on breast cancer-specific mortality. When all studies were included in the meta-analysis with each index as the unit of analysis, adherence to a high-quality diet was not significantly associated with breast cancer-specific mortality (HR = 0.91, 95% CI = 0.80, 1.02, I^2^ = 47.2%, *n* = 22). When pre-diagnosis diet quality scores and post-diagnosis diet quality scores were evaluated separately, the results were both nonsignificant; however, post-diagnosis diet quality showed a potentially larger effect (HR = 0.89, 95% CI = 0.77, 1.02, I^2^ = 51.2%, *n* = 19) than pre-diagnosis diet quality (HR = 0.98, 95% CI = 0.81, 1.17, I^2^ = 0.0%, *n* = 3) as shown in Figure 4. The results having each cohort as the unit of analysis showed a similar effect size to one from the index-based analysis: pre-diagnosis diet (HR = 0.97, 95% CI = 0.81, 1.17, I^2^ = 0.0%, *n* = 3) and post-diagnosis diet (HR = 0.85, 95% CI = 0.62, 1.18, I^2^ = 61.7%, *n* = 7) as shown in Figure 3b.

In addition, Sun et al. study [15] reported the effect of a change in diet quality. An increase in diet quality after a cancer diagnosis had an insignificant effect on reducing mortality risk; however, a decrease in diet quality showed a 67% significantly increased risk for breast cancer-specific death (HR = 1.23, 95% CI = 1.10, 2.54).

### 3.5. Subgroup Analyses by Diet Quality Index

Figure 2 and Figure 4 also demonstrate the summary effects of each post-diagnosis diet quality index on all-cause mortality and breast cancer-specific mortality, respectively.

American Cancer Society (ACS) score: Two studies [36,37] evaluated adherence scores to ACS dietary guidelines, and the results showed that higher adherence scores to ACS guidelines were not significantly associated with either all-cause mortality or breast cancer-specific mortality.

Chinese Food Pagoda (CHFP) score: Only one study [35] from the Chinese cohort used the two versions of the CHFP index, and the meta-analysis showed a significant association with reduced all-cause mortality (HR = 0.70, 95% CI = 0.57, 0.87) and breast cancer mortality (HR = 0.64, 95% CI = 0.49, 0.83).

Diet Approaches to Stop Hypertension (DASH) score: Three studies [27,35,36] evaluated the effect of adherence to DASH guidelines, and meta-analysis showed a significant inverse association between higher adherence to DASH guidelines and both all-cause mortality (HR = 0.74, 95% CI = 0.64, 0.87, I^2^ = 0.0%, *n* = 3) and breast cancer-specific mortality (HR = 0.79, 95% CI = 0.63, 0.99, I^2^ = 12.4%, *n* = 3) with evidence of little heterogeneity.

Healthy Eating Index (HEI)/Alternative HEI (AHEI): Overall, a higher HEI/AHEI score was associated with a reduced risk for all-cause mortality (HR = 0.76, 95% CI = 0.69, 0.85, I^2^ = 0.0%, *n* = 9) and for breast cancer-specific mortality (HR = 0.94, 95% CI = 0.74, 1.20, I^2^ = 48.9%, *n* = 8). More heterogeneity was observed than other diet quality indices, and only two studies [26,32] showed a statistically significant reduction in breast cancer mortality with a high score of HEI/AHEI; however, these studies had a low weight in the meta-analysis due to the small sample size. When HEI was evaluated alone [15,26,28,32,33,35,36], the results remained similar for all-cause mortality (HR = 0.78, 95% CI = 0.54, 1.12, I^2^ = 48.5%, *n* = 6, not shown in Figure 2) and breast cancer-specific mortality (HR = 0.83, 95% CI = 0.63, 1.09, I^2^ = 36.4%, *n* = 6, not shown in Figure 2).

Mediterranean Diet Score (MDS): Three studies [31,34,36] included the MDS, and the meta-analysis showed a significantly reduced risk for all-cause mortality (HR = 0.82, 95% CI = 0.68, 0.98) and a reduced but not significant risk for breast cancer-specific mortality (HR = 0.94, 95% CI = 0.65, 1.35).

### 3.6. Subgroup Analyses by Patient and Tumor Characteristics

A total of six studies [26,27,28,34,35,36] reported the subgroup analysis by the patient and tumor characteristics using the ACS score, [36] MDS, [34,36] DASH score, [27,35,36], and HEI [25,26,28,36]. The most frequently assessed variables were ER status, age or menopausal status, physical activity, and BMI. Effects of a healthy diet on all-cause mortality (Table 4) were only significant for those with BMI < 25 kg/m^2^, of postmenopausal status, and those with ER-positive, progesterone receptor (PR)-negative, or human epidermal growth factor receptor (HER) 2-positive tumors compared to their respective counterparts. Table 5 shows comparable results for breast cancer-specific mortality.

### 3.7. Sensitivity Analyses and Publication Bias

Sensitivity analysis showed that removing one of each included study in the meta-analysis did not have a significant impact on the estimated overall effect size (data not shown). A funnel plot for the analysis between the post-diagnosis diet quality and all-cause mortality showed substantial evidence of missing studies with a negative result from a small sample size (Egger test, *p* = 0.006).

### 3.8. Quality of Evidence

According to GRADE tool, the quality of evidence for the associations between pre-diagnosis diet quality and all-cause and breast cancer mortality was ‘very low’ and ‘low’, respectively (Appendix A). The quality of evidence for the associations between post-diagnosis diet quality and all-cause and breast cancer mortality was ‘low’ and ‘very low,’ due to imprecision of effect estimation, inconsistent results, and publication bias.

## 4. Discussion

To the best of our knowledge, the current study is the largest meta-analysis of cohort studies that examined the associations of diet quality scores with all-cause and cancer-specific mortality among breast cancer survivors. Our study demonstrates that better post-diagnosis diet quality could significantly reduce the risk of death from all causes by 21% and marginally reduce the risk of breast cancer death by 15%. Among the diet quality indices evaluated, post-diagnostic adherence to MDS, HEI, DASH, and CHFP, and adherence to DASH and CHFP showed significant effects on all-cause mortality and breast cancer mortality, respectively.

Our findings suggest that better diet quality after a breast cancer diagnosis can improve overall and cancer-specific survival. These results are consistent with other meta-analysis results showing reduced risks for all-cause death and cancer-specific death by 12–28% among cancer survivors [24] and noncancer older adults [42]. The biological mechanisms for diet quality and cancer outcomes are not clearly understood. Some breast cancer therapies can increase the risk of cardiovascular disease or ischemic heart disease, which could be reduced with a healthy lifestyle, including high-quality diets [43]. In addition, breast cancer survivors with better diet quality reported better quality of life, including higher mental and physical functioning scores [44], which could have a positive impact on their overall survival.

Among the 11 studies included, only three studies examined breast cancer recurrence as the outcome variable. Ergas et al. [36] reported no significant association between diet quality scores and the risk of breast cancer recurrence regardless of the diet quality index used. Wang et al. [35] reported the statistically significant effect of high diet quality scores on a lower risk of breast cancer events; however, the authors did not report the effect size for recurrence, which prevented a further meta-analysis. Therefore, future research should include breast cancer recurrence as one of the primary outcomes and report it separately.

Among the diet quality indices included, CHFP and DASH showed the most favorable results for breast cancer-specific mortality. The DASH guideline encourages high intakes of magnesium, potassium, calcium, fiber, and healthy unsaturated fats through grains, fish, and nuts and limited intake of saturated fats from red and processed meats and dairy products in addition to lower intakes of sodium. Higher fiber intakes from grains can help maintain healthy gut microbiota [45], which enhances the immune responses [46]. Park et al. [47] showed the beneficial effect of fiber on total and cause-specific mortality from a large American prospective cohort study (*n* =219,123 men and 168,999 women). While the CHFP dietary guidelines showed a significant inverse relationship to all-cause and breast cancer-specific mortality, interpretation should be made with caution due to limited data from only one study in China. Both versions of CHFP (CHFP-2007 and CHFP-2016) guidelines encourage higher intakes of vegetables, fruits, grains, and fish along with lower intakes of meat and poultry. It should be noted that most of the diet quality indices included in the current review are based on dietary guidelines for general health rather than cancer-specific health, and these guidelines encourage consuming foods or dietary components high in antioxidants, which help regulate inflammation and immune responses. Diets high in anti-inflammatory properties showed promising results in reducing the risk of breast cancer mortality [48,49,50].

We identified that better diet quality might have a more significant impact on specific groups of women, such as older women, those with a high BMI, and those with ER-positive tumors. Breast cancers found in older women are more likely to be ER-positive, and they are more likely to die from other causes than breast cancer. Moreover, those with a high BMI have an increased risk of dying from other causes, possibly due to other comorbidities. Therefore, management of comorbidities through a healthy diet can be an excellent strategy to reduce the risk of death among older cancer survivors with ER-positive tumors or a high BMI. In addition, two studies [26,35] showed that better diet quality might synergize with increased physical activity, which has been evidenced by interventional studies that used both diet and exercise interventions for weight loss among breast cancer survivors [51,52]. Thus, including both healthy diets and exercises in post-diagnostic interventions may lead to better outcomes in breast cancer survivors.

Although we focused on the relationship between diet quality and cancer outcomes, it is also important to consider other factors that influence diet quality among cancer survivors, including age, gender, education, and income levels. Individuals with low socioeconomic status (SES) are less likely to adhere to the healthy dietary guidelines [17] due to many reasons, including food insecurity [53], a lack of consistent access to enough food, and poorer access to grocery stores with a wide range of healthy foods including fresh produce [54]. Therefore, future research should address barriers preventing cancer survivors with low SES from adhering to healthy dietary guidelines. More nutrition intervention programs targeting this population are warranted to improve disease prognosis and overall health.

Inconsistent results across the studies may be explained by the differences in the study designs, including methods, timing, and frequency of dietary assessment and follow-up period in the included studies. Although FFQs are a convenient, inexpensive way to measure dietary intake in a large population-based study, it is prone to recall bias and measurement errors depending on the food items included in the questionnaire [55]. While 24-h recalls give detailed, short-term dietary intake, limitations include difficulties assessing usual, habitual intakes from a large population due to an increased burden to participants and researchers. Additionally, the timing of dietary assessment, either pre-or post-diagnosis, can influence study results. Dietary intakes after a cancer diagnosis may have different effects on short- and long-term outcomes. Deaths occurring shortly after a cancer diagnosis may be heavily influenced by cancer biology and cancer treatments rather than dietary intake. Dietary intake during cancer treatments is more variable and difficult to assess accurately. To overcome these limitations, Wang et al. [35] measured diet quality at the 5-year post-diagnosis follow-up from breast cancer survivors and counted outcome events occurring, showing the most considerable effect on all-cause mortality and breast cancer-related mortality among all studies included in the current review.

Most studies assessed diet intake at a single point in time, assuming consistent dietary intake over time within individuals, except for two studies [27,31] that used data from the Nurses’ Health Study, which collected dietary intake every four years. Kim et al. [31] and Izano et al. [27] used dietary data measured at least 12 months after diagnosis; however, the former did not use cumulative averages to avoid potential bias relating to possible changes in dietary intake resulting from a recurrence or disease progression. The cumulative average over the study periods could better represent usual intakes over the survivorship period than one-time evaluation at baseline; however, the results from the two studies were similar [27].

Only a few studies assessed the effect of individual dietary components on cancer outcomes, but with little evidence. This finding is consistent with the results from the most recent WCRF/AICR update, which stated that none of the individual dietary components were convincing or probable for reducing breast cancer-specific mortality or death from other causes [23]. The review team concluded that there was not enough evidence to make specific dietary recommendations for breast cancer survivors [23]. The WCRF/AICR update and our systematic review underscore the importance of the ‘total diet’ rather than the ‘individual nutrient/dietary component’ approach in promoting better diet quality among breast cancer survivors.

There are several limitations of the current analysis that should be considered. First, although we took the estimate from the fully adjusted multivariable model, other unmeasured confounders could not be ruled out as treatment and disease severity were not included in some studies. Second, most of the studies were conducted in the United States, with most of the sample population being white; thus, the estimates may not be directly applicable to other populations where they have different dietary intakes and risks of death. Third, included studies were all observational cohort studies; therefore, future randomized clinical trials are needed to provide more robust evidence for a relationship between healthy eating and breast cancer outcomes. Fourth, the possibility of publication bias should be considered because the small studies with null/negative results were less likely to be published. In addition, reports from theses, dissertations, and conference abstracts were not included in the current analysis. Fifth, few studies examined breast cancer recurrence as the main outcome. More research is needed to improve the body of evidence on the relationship between healthy eating and breast cancer outcomes.

There are several strengths of the current analysis worth noting. First, this is the most extensive meta-analysis of cohort studies, including 11 publications from eight cohorts, which provides greater statistical power. Another strength includes employing a cohort-based analysis approach, which allows the inclusion of all available studies but avoids over- or under-representation of a singular study by counting each study only one time. Previous meta-analysis studies counted one particular study multiple times if it included multiple dietary indices or excluded one when it was based on the same cohort as the other. Third, the current analysis included the prospective cohort studies with high quality, increasing the quality of evidence. Lastly, our study focused on a diet quality index/score that is based on a priori-defined healthy dietary recommendation rather than data-driven dietary patterns. These predefined dietary recommendations help cancer survivors easily understand how to improve their diet quality.

## 5. Conclusions

Our research demonstrates that following a healthy dietary recommendation after a cancer diagnosis can reduce mortality from all causes, including breast cancer. Additional research in large populations with different racial/ethnic backgrounds is needed to confirm our findings regarding the observed beneficial effect of DASH guidelines on breast cancer-specific mortality. It is important to increase the awareness of and compliance with healthy dietary recommendations that target overall health among cancer survivors, given that there are still no dietary guidelines specific to them. Encouraging them to adopt other healthy lifestyles, such as being physically active, avoiding drinking, and stopping smoking, will maximize the benefit of a healthy diet among breast cancer survivors.

## Figures and Tables

**Figure 1 ijerph-19-07579-f001:**
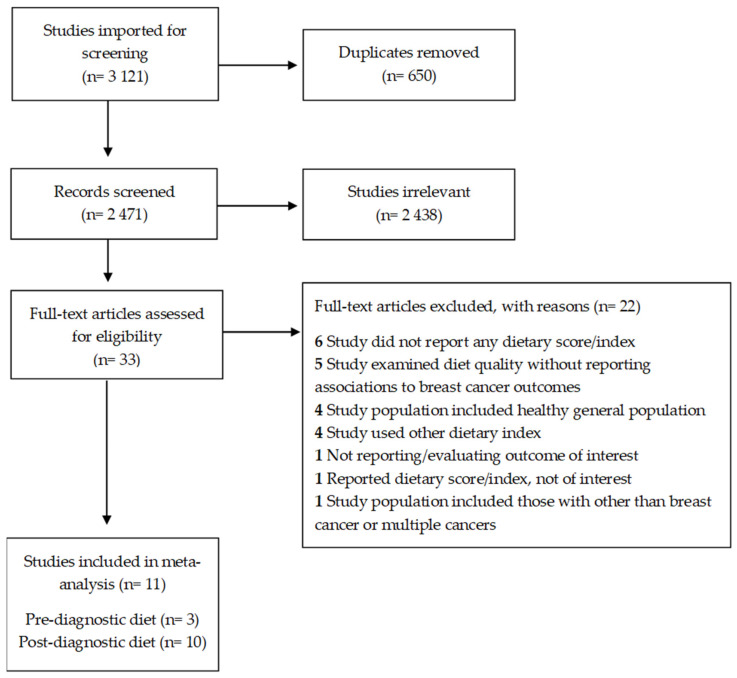
Preferred reporting items for systematic reviews and meta-analyses (PRISMA) flow chart.

**Figure 2 ijerph-19-07579-f002:**
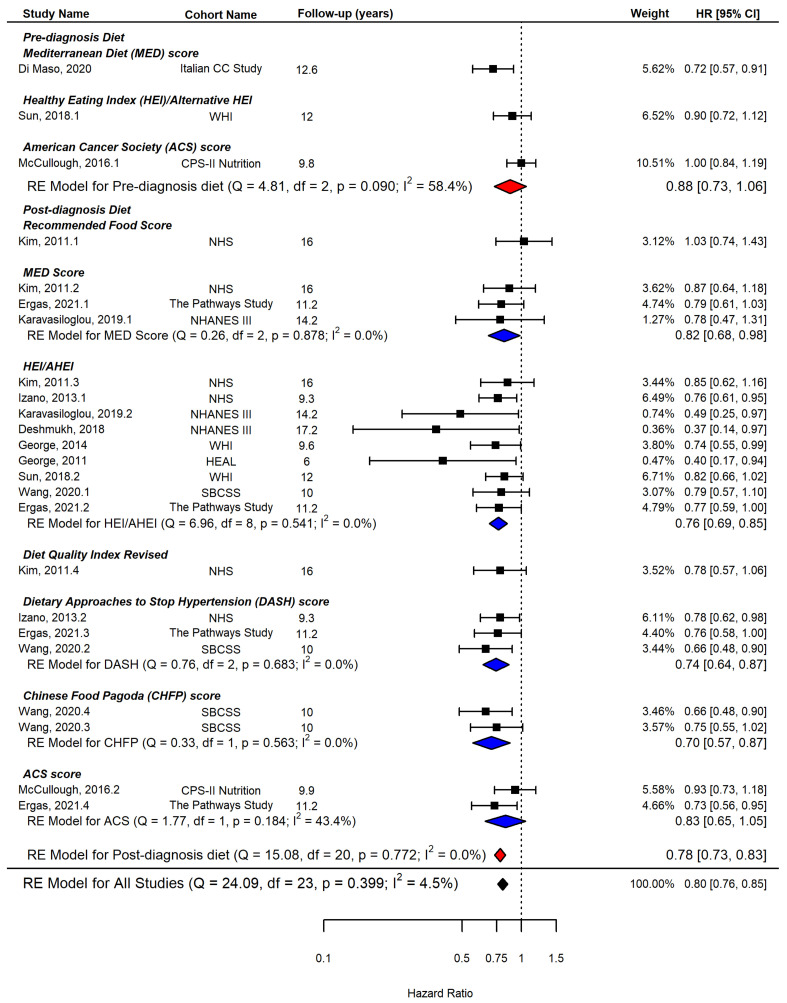
Forest plot showing pooled hazard ratios (HRs) with 95% confidence interval (CI) for association between highest vs. lowest diet quality and risk of all-cause mortality in cohort studies, diet quality index as the unit of analysis. ACS: American Cancer Society, AHEI: Alternate Healthy Eating Index, CHFP: Chinese Food Pagoda, DASH: Dietary Approaches to Stop Hypertension, DQIR: Diet Quality Index Revised, HEI: Healthy Eating Index, MDS: Mediterranean Diet Score, RFS: Recommended Food Score, CC: case–control, CPS: Cancer Prevention Study, HEAL: Health, Eating, Activity, and Lifestyle, NHANES: National Health and Nutrition Examination Survey, NHS: Nurses’ Health Study, WHI: Women’s Health Initiative, SBCSS: Shanghai Breast Cancer Survival Study [15,26,27,28,31,32,33,34,35,36,37].

**Figure 3 ijerph-19-07579-f003:**
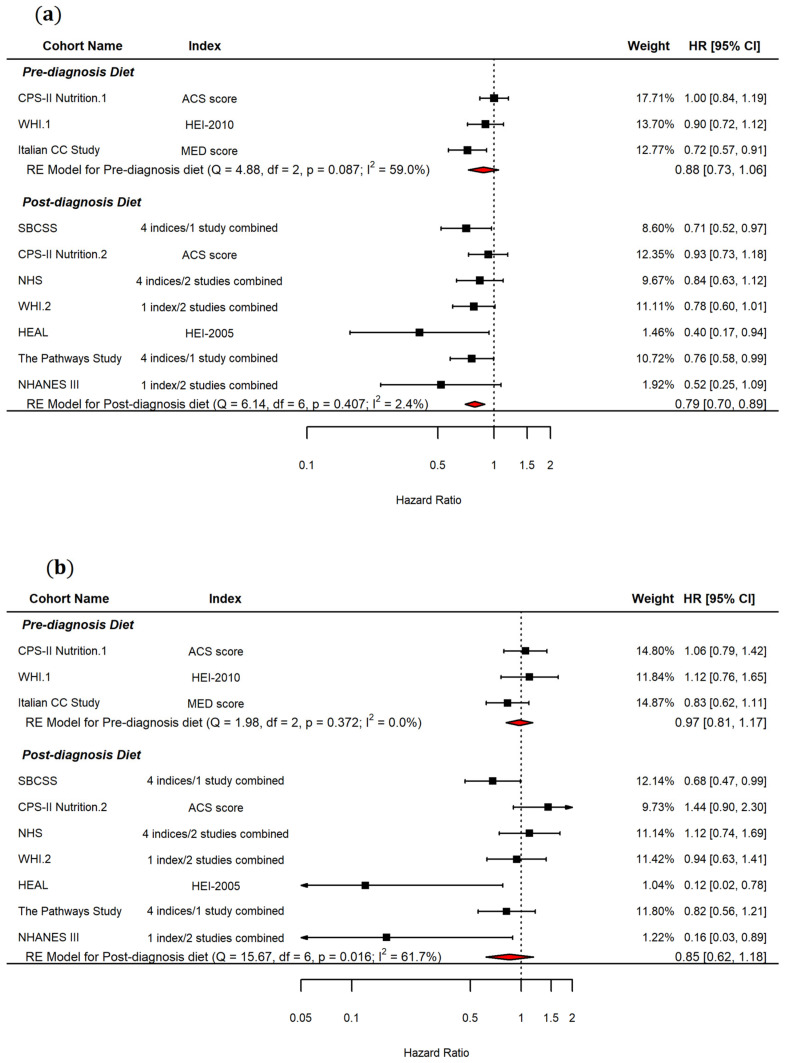
Forest plot showing pooled hazard ratios (HRs) with 95% confidence interval (CI) for association between highest vs. lowest diet quality and risk of all-cause mortality (**a**) and breast cancer-specific mortality (**b**) in cohort study, cohort as the unit of analysis. ACS: American Cancer Society, HEI: Healthy Eating Index, MED: Mediterranean Diet, CC: case–control, CPS: Cancer Prevention Study, HEAL: Health, Eating, Activity, and Lifestyle, NHANES: National Health and Nutrition Examination Survey, NHS: Nurses’ Health Study, WHI: Women’s Health Initiative, SBCSS: Shanghai Breast Cancer Survival Study.

**Figure 4 ijerph-19-07579-f004:**
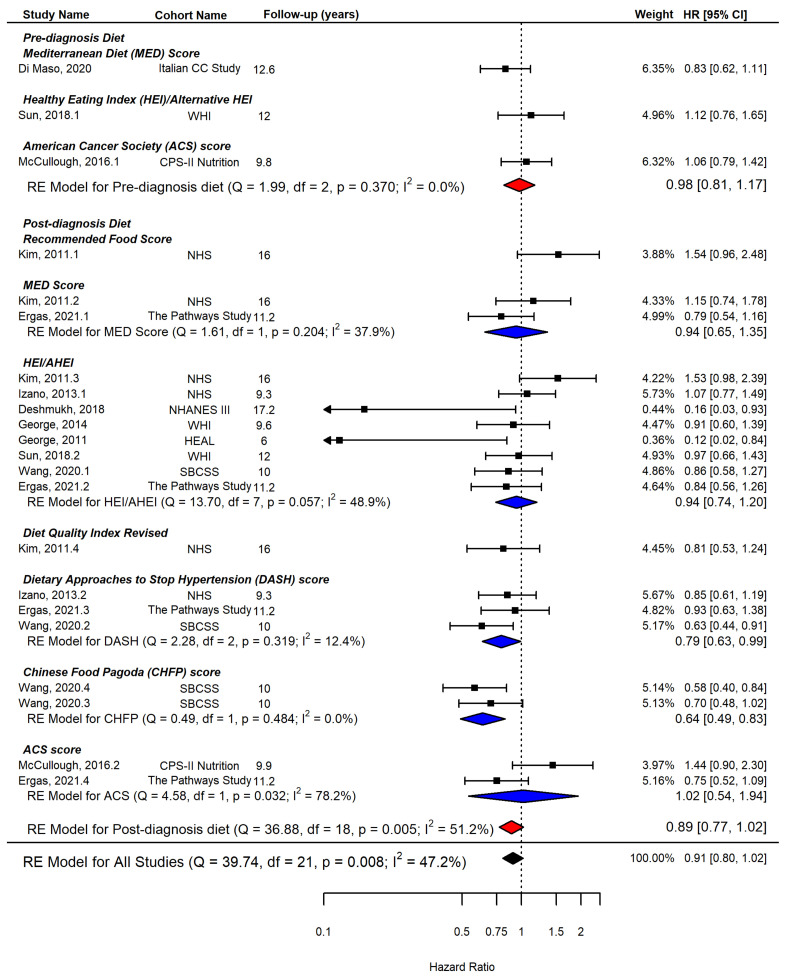
Forest plot showing pooled hazard ratios (HRs) with 95% confidence interval (CI) for association between highest vs. lowest diet quality and risk of breast cancer-specific mortality in cohort studies, diet quality index as the unit of analysis. ACS: American Cancer Society; AHEI: Alternate Healthy Eating Index, CHFP: Chinese Food Pagoda, DASH: Dietary Approaches to Stop Hypertension, DQIR: Diet Quality Index Revised, HEI: Healthy Eating Index, MDS: Mediterranean Diet Score, RFS: Recommended Food Score, CC: case–control, CPS: Cancer Prevention Study, HEAL: Health, Eating, Activity, and Lifestyle, NHANES: National Health and Nutrition Examination Survey, NHS: Nurses’ Health Study, WHI: Women’s’ Health Initiative, SBCSS: Shanghai Breast Cancer Survival Study [15,26,27,28,31,32,34,35,36,37].

**Table 1 ijerph-19-07579-t001:** Criteria for inclusion and exclusion of studies.

Criteria	Description
Participants	Adult female breast cancer survivors (age ≥ 18 years)
Exposure	Diet quality score (i.e., adherence score to predefined, healthy dietary recommendations)
Comparison	Highest vs. lowest categories of diet quality index/score
Outcome	Breast cancer recurrence and/or mortality
Study Design	Cohort study. Follow-ups of a cross-sectional or case–control study are also eligible for inclusion

**Table 2 ijerph-19-07579-t002:** Characteristics of included studies (*n* =11) examining the association between dietary quality and prognosis in female breast cancer survivors.

First Author Year Country	Cohort Name Study Type Total N Mean/Median Follow-Up Duration (year)	Age (Range) (Years) White/Black (%) Mean BMI (Distribution) Postmenopause (%) ER+ (%) Current Smokers (%)	Dietary Assessment Tool/Timing/Target	Dietary Quality Index Comparison	Outcomes Reported (Cases of Outcome) Ascertainment Methods	Multivariable-Adjusted: HR (95% CI)	Covariates Included in the Model	Study Quality ^1^
Kim 2011 USA	NHS Prospective cohort 2729 BC survivors 16 years after diagnosis	30–55 NR 25.0–26.7 NR NR 13.43%	FFQ (1980: 160 items, 1984: 130 items) ≥12 months after diagnosis Post-diagnosis diet	AHEI, DQIR, RFS, aMED Q5 vs. Q1	All-cause death (572) BC death (302) Family members, Postal service, National Death Index, Death certificate	*All-cause mortality* AHEI: 0.85 (0.63, 1.17) aMED: 0.87 (0.64, 1.17) DQIR: 0.78 (0.58, 1.07) RFS: 1.03 (0.74, 1.42) *BC mortality* AHEI: 1.53 (0.98, 2.39) aMED: 1.15 (0.74, 1.77) DQIR: 0.81 (0.53, 1.24) RFS: 1.54 (0.95, 2.47)	BMI, current smoker, physical activity, calories, alcohol, multivitamin use, oral contraceptives, postmenopausal hormone therapy, chemotherapy, radiation, tamoxifen, cancer stage	6
George 2011 USA	HEAL Prospective cohort 670 BC survivors 6 years after assessment	57.9 (18–64) 57.6% white, 28% black 27.4–28.6 60.9% 69.5% 12.69%	FFQ (122 items) 6–30 months after diagnosis Post-diagnosis diet	HEI-2005 Q4 vs. Q1	All-cause death (62) BC death (24) State mortality files, National Death Index,	*All-cause mortality* HEI-2005: 0.40 (0.17, 0.94) *BC mortality* HEI-2005: 0.12 (0.02, 0.99)	Age, race/ethnicity, menopausal status, treatment type, localized/regional, Tamoxifen use, ER status, HEI-2005 score, energy, BMI, smoking status, physical activity	6
Izano 2013 USA	NHS Prospective cohort 4103 BC survivors 112 months after diagnosis	60.4 (30–55) NR 24.9–26.9 Mix 80% 63%	FFQ ≥12 months after diagnosis, updated every 4 years Post-diagnosis diet	DASH, AHEI-2010 Q5 vs. Q1	BC death (453) Non-BC death (528) Recurrence (38) Death: Family members, Postal service National Death Index Recurrence:self-report on questionnaire	*All-cause mortality* AHEI-2010: 1.07 (0.77, 1.49) DASH: 0.85 (0.61, 1.19) *BC mortality* AHEI-2010: 1.07 (0.77, 1.49) DASH: 0.85 (0.61, 1.19) BC recurrence No associations (data not shown)	Age at diagnosis, age at first birth, parity, BMI at diagnosis, physical activity, use of oral contraceptives, postmenopausal hormones, current smoker, postmenopausal at diagnosis, ER, cancer stage, radiation treatment, chemotherapy, hormone treatment	6
George 2014 USA	WHI Prospective cohort 2317 BC survivors Median 9.6 years after assessment	63.63 (50–97) 88.6% white, 5.7% black 28.6–29.3 100% 75% NR	FFQ (122 items) 1.5 (0–6) years after diagnosis Post-diagnosis diet	HEI-2005 Q4 vs. Q1	All-cause death (415) BC death (188) National Death Index	*All-cause mortality* HEI-2005: 0.74 (0.55, 0.99) *BC mortality* HEI-2005: 0.91 (0.60, 1.40)	Age, years since diagnosis, calories, alcohol servings, MET-hours/week of MVPA, BMI, race/ethnicity, education, income, stage, ER, PR, postmenopausal hormone therapy	7
McCullough 2016 USA	CPS-II Nutrition Prospective cohort Pre: 4452 BC survivors, 9.8 years after diagnosis Post: 2152 BC survivors, 9.9 years after assessment	70.7 ± 7.2 years (40–93) 97.7% white <18.5 (0.5–1.2%) 18.5–<25 (38.6–57.8%) 25–<30 (28–34.8%) 30+ (11.1–24.5%) 79.5% 0.07%	FFQ (baseline- 68 items, follow up-152 items) 12 months after diagnosis Pre-diagnosis and post-diagnosis diet	ACS Q3 vs. Q1	Pre-diagnostic: All-cause death (1204) BC death (398) CVD death (233) Other causes of death (573) Post-diagnostic: All-cause death (640) BC death (192) CVD death (129) Other cause death (319) National Death Index	Pre-diagnostic: *All-cause mortality* ACS: 1.00 (0.84, 1.18) *BC mortality* ACS: 1.06 (0.79, 1.42) Post-diagnostic: *All-cause mortality* ACS: 0.93 (0.73, 1.18) *BC mortality* ACS: 1.44 (0.90, 2.30)	Age at diagnosis, year of BC diagnosis, race/ethnicity, tumor stage at diagnosis, tumor grade at diagnosis, ER, PR, surgery, radiation, chemotherapy as initial treatment, BMI, cigarette smoking status, physical activity, hormone replacement therapy	7
Deshmukh 2018 USA	NHANES III Retrospective cohort 131 BC survivors Median 17.2 years after assessment	(40–69) 95% white NR NR NR NR	24-h recall NR Post-diagnosis diet	HEI 1994–1996 Q4 vs. Q1	All-cause death (NR) BC death (NR) National Center for Health Statistics Linked Mortality Files	*All-cause mortality* HEI: 0.59 (0.45, 0.77) *BC mortality* HEI: 0.40 (0.18, 0.89)	Age, sex, income, education, and BMI	6
Sun 2018 USA	WHI Prospective cohort 2295 BC survivors 12 years after assessment	65.92 (50–79) 88.8% white, 5.7% black NR 100% 74.3% 5.03%	FFQ (122 items) Pre-diagnosis diet—average 1.5 years before diagnosis; post-diagnosis diet—average 1.8 years from diagnosis Pre-and post-diagnosis diet; Change in diet quality	HEI-2010 Q4 vs. Q1 Increase (≥15%) or decrease ((≥15%) vs. no change or stable (±14.9%)	All-cause death (763) BC death (242) Non-BC death (521) National Death Index	Pre-diagnosis diet *All-cause mortality* HEI-2010: 0.90 (0.72, 1.12) *BC mortality* HEI-2010: 1.12 (0.76, 1.64) Post-diagnosis diet *All-cause mortality* HEI-2010: 0.82 (0.66, 1.02) *BC mortality* HEI-2010: 0.97 (0.66, 1.43) Change of diet quality Increase *All-cause mortality* HEI-2010: 1.00 (0.81, 1.23) *BC mortality* HEI-2010: 0.98 (0.67, 1.44) Decrease *All-cause mortality* HEI-2010: 1.23 (0.99, 1.62) *BC mortality* HEI-2010: 1.67 (1.10, 2.54)	Age at diagnosis, total energy intake, race or ethnicity, education, income, breast cancer stage, ER status, PR status, smoking, physical activity, intervention arm, use of postmenopausal hormone therapy, alcohol intake, and BMI (post-diagnosis only-time from diagnosis to dietary intake assessment).	8
Karavasiloglou 2019 USA	NHANES III Retrospective cohort 110 BC survivors Median 8.6 years after assessment	62.4 91.6% white 26.4 ± 0.5 NR NR 16.9%	24-h recall NR Post-diagnosis diet	HEI (good vs. poor), MDS (adherers vs. non-adherers)	All-cause death (NR) National Death Index	*All-cause mortality* HEI: 0.49 (0.25, 0.97) MDS: 0.78 (0.47, 1.32)	Age at survey, age at diagnosis, time from the completion of the questionnaire until the end of the follow-up, race/ethnicity, marital status, SES status, smoking status, physical activity, BMI, daily energy intake, history of menopausal hormone therapy use, prevalent chronic diseases at baseline	7
Wang 2020 China	SBCSS Prospective cohort 3450 invasive BC survivors 8 years after assessment	25–70 NR 24.0 ± 3.3–24.6 ± 3.8 49.57% 65.6% NR	FFQ (93 items) 5 years after surgery Post-diagnosis diet	CHFP-2007, CHFP-2016, DASH, HEI-2015 Q4 vs. Q1	All-cause death (374) BC death (252) Non-BC death (122) BC events ^2^ (228) Shanghai Vital Statistics Registry	*All-cause mortality* CHFP-2007: 0.66 (0.48–0.89) CHFP-2016: 0.75 (0.55–1.01) mDASH: 0.66 (0.49–0.91) HEI-2015: 0.79 (0.57–1.10) *BC mortality* CHFP-2007: 0.58 (0.40, 0.84) CHFP-2016: 0.70 (0.48, 1.01) mDASH: 0.63 (0.44, 0.92) HEI-2015: 0.86 (0.58, 1.27) *BC events* ^2^ CHFP-2007: 0.84 (0.74–0.95) CHFP-2016: 0.84 (0.74–0.95) mDASH: 0.92 (0.85–0.99) HEI-2015: 0.92 (0.81- 1.05)	Age at dietary survey, interval between diagnosis and dietary survey, and total energy intake, income, education, marriage, menopausal status, BMI, physical activity, ER status, PR status, HER2 status, TNM stage, comorbidity, chemotherapy, radiation, and immunotherapy	8
DiMaso 2020 Italy	Italian Case–Control Study Retrospective cohort 1453 BC survivors Median 12.6 years after diagnosis	55 (23–78) NR <25 (21.9–36.1%) 25–29.9 (23.7–33.4%) ≥30 (25.6–29.1%) 62% NR 19.96%	FFQ (78 items) 2 years prior to diagnosis Pre-diagnosis diet	MDS Q3 vs. Q1	All-cause death (503) BC death (365) Non-BC death (138) Population-based regional cancer registries	*All-cause mortality* MDS: 0.72 (0.57, 0.92) *BC mortality* MDS: 0.83 (0.62, 1.11)	Study design variables (area of residence, calendar period of cancer diagnosis), socio-demographic characteristics (age at diagnosis, education, menopausal status), clinical cancer features (TNM stage, ER/PR status), and total energy intake.	8
Ergas 2021 USA	The Pathways Study Prospective cohort 3660 BC survivors 40,888 person-years	9.7 (24–94) 68% white, 6.6% black 26.3–29.9 71% 83.96% 4.2%	FFQ (139 items) 2.3 months (range = 0.7–18.7) after diagnosis Post-diagnosis diet	ACS, aMED, DASH, HEI-2015 Q5 vs. Q1	All-cause death (655) BC death (324) BC recurrence (461) Non-BC death (331) Follow-up interviews with relatives of participants, Medical chart review, Linkages with data from the state of California Social Security Administration, National Death Index	*All-cause mortality* ACS: 0.73 (0.56, 0.95) aMED: 0.79 (0.61, 1.03) DASH: 0.76 (0.58, 1.00) HEI-2015: 0.77 (0.6, 1.01) *BC mortality* ACS: 0.75 (0.52, 1.09) aMED: 0.79 (0.54, 1.16) DASH: 0.93 (0.63, 1.39) HEI-2105: 0.84 (0.56, 1.27) *BC recurrence* ACS: 1.19 (0.89, 1.57) aMED 1.08 (0.79, 1.47) DASH: 1.02 (0.73, 1.41) HEI-2015: 1.24 (0.88, 1.75)	Age at diagnosis and total energy, race and ethnicity, education level, menopausal status, physical activity, smoking, cancer stage, ER, PR, HER2, BMI, type of surgery, chemotherapy, radiation, and hormonal therapies.	8

^1^ A summary score was calculated using the Newcastle–Ottawa Scale for cohort studies, and studies that received a score of 6 or above were considered high quality. ^2^ BC events including recurrence/metastasis or breast cancer-specific mortality. Participants who reported breast cancer recurrence/metastasis before the dietary survey (*n* = 175) or participants who were lost to follow-up at 10-year post-diagnosis in-person follow-up survey and did not die from breast cancer (*n* =189) were excluded from breast cancer-specific events analyses, resulting in 3088 participants and 228 events. NR: not reported; BC: breast cancer, BMI: body mass index, ER: estrogen receptor, HER2: human epidermal growth factor receptor 2, PR: progesterone receptor, TNM: tumor, node, Metastasis, ACS: American Cancer Society, AHEI: Alternate Healthy Eating Index, aMED: Alternate Mediterranean Diet, CHFP: Chinese Food Pagoda, DASH: Dietary Approaches to Stop Hypertension, DQIR: Diet Quality Index Revised, HEI: Healthy Eating Index, MDS: Mediterranean Diet Score, RFS: Recommended Food Score, CPS: Cancer Prevention Study, HEAL: Health, Eating, Activity, and Lifestyle, NHANES: National Health and Nutrition Examination Survey, NHS: Nurses’ Health Study, WHI: Women’s Health Initiative, SBCSS: Shanghai Breast Cancer Survival Study.

**Table 3 ijerph-19-07579-t003:** Comparison of dietary quality indices included in the systematic review.

Diet Quality Index: Components (Score Range)	Encouraged Components (Number)	Discouraged/Moderation Components (Number)	Effect of Individual Components
HEI: 10 (0–100) HEI-2005: 12 (0–100) HEI-2010: 12 (0–100) HEI-2015: 13 (0–100)	HEI (5) vegetables, fruits, grain, dairy, variety HEI-2005 (7) total fruit, whole fruit, total vegetables, dark green and orange vegetables and legumes, total grains, whole grains, milk HEI-2010 (8) total fruit, whole fruit, total vegetables, greens and beans, whole grains, dairy, total protein foods, seafood and plant proteins HEI-2015 (8) total fruits, whole fruits, total vegetables, greens and beans, total protein, seafood and plant protein, whole grains, dairy	HEI (5) meat, fat, saturated fat, cholesterol, sodium HEI-2005 (5) meat and beans, oils, saturated fat, sodium, and calories from solid fats/alcoholic beverages/added sugars HEI-2010 (4) fatty acids (PUFAs + MUFAs)/SFAs), refined grains, sodium, empty calories HEI-2015 (5) refined grains, added sugars, fatty acids, sodium, saturated fats	Deshmukh 2018—NR Ergas 2021—decreased intake of refined grain/sodium had a lower risk; higher intake of whole grains/nuts had a higher risk of all-cause mortality George 2011—no effect George 2014—NR Sun 2018—NR Wang 2020—NR
AHEI: 9 (0–90) AHEI-2010: 11 (0–110)	AHEI (5) vegetables, fruits, nuts, soy, cereal fiber AHEI-2010 (7) vegetables, fruits, nuts and legumes, whole grains, trans fats, long-chain (*n* − 3) fats (EPA + DHA), polyunsaturated fats	AHEI (4) ratio of white to red meat, trans fat, polyunsaturated:saturated fat ratio, alcohol AHEI-2010: (4) sugar-sweetened beverages and fruit juice, red/processed meat, sodium, alcohol	Kim 2011—NR Izano 2013—NR
DASH: 8 (0–40) m-DASH: 7 (0–70)	DASH (5) fruits, vegetables, nuts, grains, low-fat dairy m-DASH (4) fruits and vegetables, dairy products, fish and eggs, nuts (nuts, beans, legumes)	DASH (3) red/processed meats, sugar-sweetened beverages, sodium m-DASH (3) poultry, fats and oil, sodium	Ergas 2021—no effect Izano 2013—NR Wang 2020—NR
ACS: 3 (0–9)	(2) total fruits and vegetables, whole grains	(1) Total red and processed meats	Ergas 2021—greater intake of whole grains had a lower risk of all-cause mortality McCullough 2016—lower red/processed meats after diagnosis had lower risk of total, CVD, and non-breast cancer mortality
MDS: 9 (0–9) aMED:9 (0–90)	MDS (6) fruit, vegetables, legumes, fish, MUFA/SFA ratio, cereal aMED (7) vegetables, legumes, fruits and nuts, whole grain, cereals, fish, MUFA/SFA ratio	MDS (3) meats, total dairy, alcohol aMED (2) red/processed meats, alcohol	DiMaso 2020—NR Ergas 2021—greater intake of nuts had a lower risk of all-cause mortality Kim 2011—NR
CHFP-2007:10 (0–45) CHFP-2016:10 (0–45)	CHFP-2007 and 2016: (7) fruits, vegetables, grains, fish, eggs, beans, dairy products	CHFP-2007 and 2016: (3) meat and poultry, fats and oil, salt	Wang 2020—NR
RFS: 5 (0–56)	(5) fruits, vegetables, whole grains, low saturated fat proteins, low fat dairy products	NR	Kim 2011—NR
DQIR: 10 (0–100)	(9) grains, vegetables, fruits, total fat, saturated fat, cholesterol, iron, calcium, diet diversity	(1) added fat and sugar moderation	Kim 2011—NR

ACS: American Cancer Society, AHEI: Alternate Healthy Eating Index, aMED: Alternate Mediterranean Diet, CHFP: Chinese Food Pagoda, DASH: Dietary Approaches to Stop Hypertension, DHA: Docosahexaenoic Acid, DQIR: Diet Quality Index Revised, EPA: eicosapentaenoic acid, HEI: Healthy Eating Index, m-DASH: modified Dietary Approaches to Stop Hypertension, MDS: Mediterranean Diet Score, MUFA: monounsaturated fatty acid, NR: not reported, PUFA: polyunsaturated fatty acid, RFS: Recommended Food Score, SFA: saturated fatty acid.

**Table 4 ijerph-19-07579-t004:** Subgroup analysis by patient and clinical characteristics for all-cause mortality comparing those in highest and lowest categories of diet quality.

		All-Cause Mortality HR (95% CI)					
	Subgroup	Ergas 2021	Di Maso 2020	Wang 2020	George 2014	George 2011	Meta-Analysis ^1^
Age	Young	-	MDS: 1.01 (0.69, 1.48)	m-DASH: 0.99 (0.88, 1.08)	-	-	0.96 (0.83, 1.10)
	Old	-	**MDS: 0.55 (0.39, 0.76)**	**m-DASH: 0.90 (0.83, 0.97)**	-	-	0.72 (0.45, 1.17)
Menopausal status	Pre	-	MDS: 1.01 (0.65, 1.58)	-	-	-	1.01 (0.65, 1.58)
	Post	-	**MDS: 0.65 (0.48, 0.87)**	-	-	-	**0.65 (0.48, 0.87)**
Body mass index	<25 kg/m^2^	-	MDS: 0.81 (0.58, 1.14)	m-DASH: 0.93 (0.85, 1.01)	-	-	**0.92 (0.85, 1.00)**
	≥25 kg/m^2^	-	**MDS: 0.64 (0.45, 0.92)**	m-DASH: 0.91 (0.83, 1.00)	-	-	0.80 (0.57, 1.11)
Physical activity	Low	-	-	m-DASH: 0.95 (0.73, 1.03)	-	HEI-2005: 1.07 (0.30, 3.84)	0.99 (0.89, 1.10)
	High	-	-	**m-DASH: 0.87 (0.79, 0.96)**	-	**HEI-2005: 0.11 (0.04, 0.36)**	0.31 (0.04, 2.35)
ER	Positive	ACS: 0.68 (0.51, 1.01) aMED: 0.75 (0.56, 1.01) **DASH: 0.70 (0.52, 0.95)** HEI-2015: 0.80 (0.60, 1.06)	-	**m-DASH: 0.93 (0.87, 1.00)**	HEI-2005: **0.55 (0.38, 0.79)**	-	**0.88 (0.82, 0.93**)
	Negative	ACS: 1.05 (0.59, 1.89) aMED: 0.92 (0.49, 1.71) DASH: 1.25 (0.64, 2.43) HEI-2015: 0.73 (0.38, 1.40)	-	m-DASH: 0.91 (0.81, 1.03)	HEI-2005: 1.14 (0.58, 2.23)	-	0.92 (0.83, 1.03)
PR	Positive	-	-	m-DASH: 0.95 (0.88, 1.02)	-	-	0.95 (0.88, 1.02)
	Negative	-	-	**m-DASH: 0.88 (0.79, 0.99)**	-	-	**0.88 (0.79, 0.99)**
HER2	Positive	-	-	**m-DASH: 0.83 (0.71, 0.88)**	-	-	**0.83 (0.71, 0.88)**
	Negative	-	-	**m-DASH: 0.90 (0.83, 0.98)**	-	-	**0.90 (0.83, 0.98)**

^1^ Results are from the random-effects model meta-analysis, and significant findings are in bold. ACS: American Cancer Society, aMED: Alternate Mediterranean Diet, AHEI: Alternative Healthy Eating Index, DASH: Dietary Approaches to Stop Hypertension, MDS: Mediterranean Diet Score, mDASH; modified Dietary Approaches to Stop Hypertension, HEI: Healthy Eating Index, HR, hazard ratio, ER: estrogen receptor, HER2: human epidermal growth factor receptor, PR: progesterone receptor.

**Table 5 ijerph-19-07579-t005:** Subgroup analysis by patient and clinical characteristics for breast cancer mortality comparing those in highest and lowest categories of diet quality.

		Breast Cancer Mortality HR (95% CI)				
	Subgroup	Wang 2020	Di Maso 2020	Izano 2013	George 2011	Meta-Analysis ^1^
Age	Young	m-DASH: 0.97 (0.87, 1.09)	MDS: 1.06 (0.69, 1.61)	-	-	0.98 (0.88, 1.09)
	Old	**m-DASH: 0.86 (0.78, 0.95)**	**MDS: 0.65 (0.43, 0.98)**	-	-	**0.81 (0.64, 1.02)**
Menopause	Pre	-	MDS: 1.06 (0.65, 1.71)	-	-	1.06 (0.65, 1.71)
	Post	-	MDS: 0.73 (0.51, 1.05)	-	-	0.73 (0.51, 1.05)
BMI	<25 kg/m^2^	-	MDS: 0.73 (0.48, 1.11)	-	-	0.73 (0.48, 1.11)
	≥25 kg/m^2^	-	MDS: 0.97 (0.64, 1.46)	-	-	0.97 (0.64, 1.46)
Physical activity	Low	m-DASH: 0.93 (0.84, 1.03)	-	-	HEI-2005: 1.88 (0.41, 8.65) ^2^	0.93 (0.84, 1.03)
	High	**m-DASH: 0.88 (0.78, 0.98)**	-	-	**HEI-2005: 0.09 (0.01, 0.89)**	0.37 (0.04, 3.26)
ER	Positive	**m-DASH: 0.91 (0.84, 1.00)**	-	AHEI-2010: 0.89 (0.30, 2.66)DASH: 0.87 (0.58, 1.32)	-	**0.88 (0.76, 1.03)**
	Negative	m-DASH: 0.89 (0.77, 1.05)	-	AHEI-2010: 0.89 (0.30, 2.66)DASH: 0.65 (0.22, 1.93)	-	0.91 (0.83, 0.99)
PR	Positive	m-DASH: 0.95 (0.87, 1.04)	-	-	-	0.95 (0.87, 1.04)
	Negative	**m-DASH: 0.83 (0.73, 0.96)**	-	-	-	**0.83 (0.73, 0.96)**
HER2	Positive	**m-DASH: 0.73 (0.60, 0.90)**	-	-	-	**0.73 (0.60, 0.90**)
	Negative	m-DASH: 0.91 (0.82, 1.02)	-	-	-	0.91 (0.82, 1.01)

^1^ Results are from the random-effects model meta-analysis, and significant findings are in bold. ^2^ Comparison is made between mixed-quality diet (Q2-Q3) and poor-quality diet (Q1) due to no observed death in better-quality diet group (Q4). ACS: American Cancer Society, AHEI: Alternative Healthy Eating Index, DASH: Dietary Approaches to Stop Hypertension, MDS: Mediterranean Diet Score, mDASH; modified Dietary Approaches to Stop Hypertension, HEI: Healthy Eating Index, HR, hazard ratio, ER: estrogen receptor, HER2: human epidermal growth factor receptor, PR: progesterone receptor.

## Data Availability

Not applicable.

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
