# Peer review of "Healthy Eating and Mortality among Breast Cancer Survivors: A Systematic Review and Meta-Analysis of Cohort Studies"

_ijerph, 2022, doi:10.3390/ijerph19137579_

Round 1
Reviewer 1 Report
It appears that the authors addressed all of my comments. good luck
Author Response
We appreciate time and energy the reviewer committed and thank you for
the positive comments.
Reviewer 2 Report
Please, see the attached document.

Author Response
Provide at least one reference that support the following sentence: “Cancer survivors tendto change their food choices following a cancer diagnosis, hoping to influence their prognosis positively”.
• RESPONSE: Thank you for your suggestion. We inserted one reference #2 in line 39.
Beeken, R. J., Williams, K., Wardle, J., & Croker, H. (2016). “What about diet?” A
qualitative study of cancer survivors' views on diet and cancer and their sources of information. European Journal of Cancer Care, 25(5), 774–783.
https://doi.org/10.1111/ecc.12529. This article reports ““Participants did occasionally mention that they had made dietary changes to avoid cancer recurrence”.
Revise the sentence “Many epidemiologic studies have shown that diet quality, such as Healthy Eating Index (HEI), is strongly and positively associated with obesity and a higher body mass index (BMI)…”. Specify the level of diet quality that “is strongly and positively associated with obesity and a higher body mass index (BMI)”.
RESPONSE: We agree with you and specified the level of diet quality such that ‘poor diet quality is strongly and positively associated with obesity and a higher body mass index’ in line 40.
I am not convinced that poor diet quality is associated with breast density, having as justification the reference # 8 (Castelló et al., 2015). Please, include another reference to justify such observation. In fact, “women with less body fat are more likely to have more dense breast tissue compared with women who are obese”
(https://www.mayoclinic.org/tests-procedures/mammogram/in-depth/dense-breasttissue/art-20123968)
• RESPONSE: Thank you for this suggestion. We included two references (#10, 11 in line 44) showing how breast density is influenced by diet. 1) Jones, J. A., Hartman, T. J., Klifa, C. S., Coffman, D. L., Mitchell, D. C., Vernarelli, J. A., Snetselaar, L. G., Van Horn, L., Stevens, V. J., Robson, A. M., Himes, J. H., Shepherd, J. A., & Dorgan, J. F. (2015). Dietary energy density is positively associated with breast density among young women. Journal of the Academy of Nutrition and Dietetics, 115(3), 353–359.
https://doi.org/10.1016/j.jand.2014.08.015. Please note that dietary energy density is a measure of diet quality.
2) Masala G, Ambrogetti D, Assedi M, Giorgi D, Del Turco MR, Palli D. Dietary and
lifestyle determinants of mammographic breast density. A longitudinal study in a
Mediterranean population. Int J Cancer 2006;118:1782–9.
6. Revise the sentence “… and breast density, [8] which are associated with cancer growth”. Up to the reviewer’s knowledge breast density is a risk factor for breast cancer since it makes difficult the detection of breast cancer malignancies; breast density is not associated to cancer growth. Please see, https://www.cancer.gov/types/breast/breast-changes/densebreasts#:~:text=How%20common%20are%20dense%20breasts,having%20children%2C%20and%20using%20tamoxifen
• RESPONSE: Thank you for this suggestion, which helps clarification. We have replaced the word ‘cancer growth’ with ‘cancer risk’ in line 44.
The sentence “Another mechanism of a healthy diet that can lead to better cancer survival could include controlling tumor promotors through improved weight loss and insulin sensitivity with a better diet” is out of context. Please, see the section “weight gain and obesity” at https://www.wcrf.org/diet-activity-and-cancer/global-cancerupdateprogramme/cancersurvivors/#:~:text=eat%20more%20wholegrains%2C%20vegetables%2C%20fruits,avoid%20processed%20meats%20and%20alcohol
• RESPONSE: We agree with you that weight gain after a diagnosis or a high body mass index at cancer diagnosis may be associated with longer survival (obesity paradox) in some diseases. However, please note that the sentence was not created by the authors, and we paraphrased the conclusion of the Blackburn’s study, which showed that an intervention with a low-fat diet (i.e., a better diet) improved survival through a weight loss and improved insulin resistance.
• We have added another reference #14 below that supports this claim in line 47.
Jackson, S. E., Heinrich, M., Beeken, R. J., & Wardle, J. (2017). Weight Loss and Mortality in Overweight and Obese Cancer Survivors: A Systematic Review. PloS one, 12(1), e0169173. https://doi.org/10.1371/journal.pone.0169173
The content of the citation “Sun et al. [11] showed that 28% of breast cancer survivors changed their diet quality after a cancer diagnosis” could not be found in the reference #11
• RESPONSE: This is a valid assessment because Sun et al. did not explicitly show the percentage of the sample that made a change in their diet quality; however, we believe that 28% is a correct number. In the text, it showed that 72% of the women maintained relatively stable diet quality, 9% decreased diet quality, and 19% increased diet quality. Thus, we summarized and paraphrased it to ‘28% (9% + 19%) changed their diet quality’.
In the sentence “Sun et al. [11] showed that 28% of breast cancer survivors changed their diet quality after a cancer diagnosis, and Thompson et al. [12] reported increased vegetable and fruit consumption and decreased dietary fat consumption after a cancer diagnosis” specify the study population. The availability and accessibility to vegetables and fruits can vary by characteristics such as geographical location and socioeconomic status.
• RESPONSE: We appreciate this valuable comment. We have specified the population in lines 48-52.
• Sun – breast cancer survivors who participated in the WHI study in the United States.
• Thompson – women treated for invasive breast cancer in the United States.
After the sentence “One reason behind this effect heterogeneity may be that there are no dietary guidelines specifically for cancer survivors”, add the citation for: Rock, C. L., Thomson, C. A., Sullivan, K. R., Howe, C. L., Kushi, L. H., Caan, B. J., Neuhouser, M. L., Bandera, E. V., Wang, Y., Robien, K., Basen-Engquist, K. M., Brown, J. C., Courneya, K. S., Crane, T. E., Garcia, D. O., Grant, B. L., Hamilton, K. K., Hartman, S. J., Kenfield, S. A., Martinez, M. E., … McCullough, M. L. (2022). American Cancer Society nutrition and physical activity guideline for cancer survivors. CA: a cancer journal for clinicians, 72(3), 230–262.
https://doi.org/10.3322/caac.21719 (https://acsjournals.onlinelibrary.wiley.com/doi/10.3322/caac.21719).
RESPONSE: Thank you for this suggestion. We added reference #19 in line 59.
Please, make more evident the reasons why this meta-analysis is necessary in the following paragraph: “Employing a cohort-based meta-analysis rather than an index-based analysis will produce a more valid estimate of these measures. In addition, including all available studies will provide sufficient statistical power to identify a diet quality index that has the most favorable impact on breast cancer outcomes, and subgroup analysis will identify target populations for interventions”. What does this manuscript add to the previous published meta-analyses?
• RESPONSE: Thank you for this suggestion; however, we believe that we explained the reasons why this meta-analysis employing a cohort-based analysis is necessary referencing to the previous published meta-analyses’ limitations in the previous paragraph starting ‘However, they counted one study multiple times ~ ~ ~ could lead to a biased summary measure. Special attention is needed for meta-analysis when multiple dietary indices are evaluated in one study or when multiple publications are produced from the same cohort.’ The sentence of “employing a cohort-based meta-analysis ~ ~” describes potential benefits that we could expect from this approach. We have retained the sentences.
The aims specified in the introduction are: (1) to examine the associations between dietquality indices/scores and mortality outcomes using two approaches: i) a dietary index as the unit of analysis (index-based analysis) and ii) a cohort as the unit of analysis (cohortbased analysis) and 2) to examine whether such associations vary according to cancer subtype or participant characteristics”. They do not match with “the identification of cohort studies that examined the associations between diet quality and breast cancer recurrence and mortality” specified in the methods section
• RESPONSE: Thank you for your insightful observation. We agree with you and have revised the aims in the introduction (lines 97-98) to match with ones in the methods section.
Did the Korean and Spanish studies were translated to English? Explain the process.
• RESPONSE: [This is an interesting query. As we explained in the first revision, all
abstracts were available in English and none of the Korean and Spanish studies were included in the full-text review stage, so we did not need to translate them although we planned to translate ones in Korean and Spanish to in English. We have added the following information in lines 119-122. ‘Abstracts were available in English although the full texts were in Korean or Spanish. Translation of Korean or Spanish to English was planned when the publication was to pass the initial title/abstract screening phase, but none passed the screening’.
Although the authors specified that the review was restricted to cohort studies, they included additional study designs. For example, they included the “Italian Case-Control Study” and classified it as a retrospective cohort. Also, they included the “NHANES III” (a panel study) and classified it as a retrospective cohort. Please, restrict your study to concurrent and non-concurrent cohort studies. A cross-sectional study with a follow-up is classified as a panel study. A case-control study starts with the definition of the event or outcome.
• RESPONSE: Please see point 1 above.
Reviewer 3 Report
Not all of my comments have been included in the repairs, but I accept publication in this form.
Author Response
Not all of my comments have been included in the repairs, but I accept publication in this form.
• We are sorry that our revision is not at your satisfaction although we tried our best. We appreciate your time to review the revision.
This manuscript is a resubmission of an earlier submission. The following is a list of the peer review reports and author responses from that submission.
Round 1
Reviewer 1 Report
This study presents exciting research on an important topic. Very well written. It would contribute to the literature on diet quality of health outcomes. I have a few concerns and comments before it can be considered for publication. Please see below.
Line 20: Spell out, ER-positive, PR-negative, or HER2-positive tumors, I2, df(Q)
Line 24: Please replace “;” with “,” before meta-analysis
Replace “diet quality score” with “Healthy Eating Index (HEI)-2015 score.”
The introduction is not sufficient to motivate the topic of interest. Since the diet quality index itself does not affect diseases, development authors could emphasize the mechanism through which they are interconnected. For example, previous studies have shown that diet quality, such as Healthy Eating Index, is strongly and positively associated with obesity and higher BMI, a major risk factor for several non-communicable diseases. The authors could give more background information on the association between diet quality and health outcomes in this context. This could help to shed light on the potential mechanism through which diet quality affects the incidence of cancer and cancer-related deaths. Yet, this section is entirely missing in the introduction. Through my google scholar search, I found a few articles related to this:
- Dhakal, C.K.; Khadka, S. Heterogeneities in Consumer Diet Quality and Health Outcomes of Consumers by Store Choice and Income. Nutrients2021, 13, 1046.
- Saraiya, V., Bradshaw, P., Meyer, K., Gammon, M., Slade, G., Brennan, P., ... & Olshan, A. (2020). The association between diet quality and cancer incidence of the head and neck. Cancer Causes & Control, 31(2), 193-202.
- GBD 2015 Obesity Collaborators. (2017). Health effects of overweight and obesity in 195 countries over 25 years. New England Journal of Medicine, 377(1), 13-27.
Also, it is well known that individuals with low socioeconomic status (SES) are more likely to eat a poor diet and develop diseases. Although outside the scope of this study, you can give a little bit of background info on the differential effects across SES.
Font size and type is figure are not consistent with the text.
Please make sure the text inside figure 2 is consistent with the text and stand-alone.
The discussion is too long. Please make it concise.
Would you please check the reference style and make sure it follows the journal's guidelines?
Reviewer 2 Report
Although the authors spent a lot of time collecting and comparing data from individual studies, for me as a scientist, these results have no value and I did not learn anything from them that I did not know before.
In the materials and methods, I read only how the selected studies were compared, but I have no idea how the results were obtained in these studies. So, to find out, I still have to open individual publications. I lack knowledge of whether the data were obtained using an application, whether they were in the form of questionnaires obtained during an ambulance visit, whether the patients were called by phone .....Although these findings are briefly outlined in the discussion (line 354-358), it would be desirable to refine these data for each study observed and to specify them in the methods.
What surprised me was the knowledge about diet 2 to 9 years before the diagnosis. Does this mean that patients were randomly monitored or how was this data recovered?
Very important findings are in Table 4, but it would be good to remake this table and compare patients ER positive, PR positive, HER2 positive .... and the effect of 7 different diet quality indices / scores measuring. After all, it is very important whether a change in diet also has an effect on genetically determined cancers.
In the discussion (line 304, 309,310) you describe the effect of proinflammatory cytokines, IGF-1 or insulin sensitivity, but in your work such findings are missing.
Reviewer 3 Report
Please, see the attached file.
